# Enhancing Lung Cancer Classification through Integration of Liquid Biopsy Multi-Omics Data with Machine Learning Techniques

**DOI:** 10.3390/cancers15184556

**Published:** 2023-09-14

**Authors:** Hyuk-Jung Kwon, Ui-Hyun Park, Chul Jun Goh, Dabin Park, Yu Gyeong Lim, Isaac Kise Lee, Woo-Jung Do, Kyoung Joo Lee, Hyojung Kim, Seon-Young Yun, Joungsu Joo, Na Young Min, Sunghoon Lee, Sang-Won Um, Min-Seob Lee

**Affiliations:** 1Eone-Diagnomics Genome Center, Inc., 143, Gaetbeol-ro, Yeonsu-gu, Incheon 21999, Republic of Korea; hjkwon@edgc.com (H.-J.K.); uh.park@edgc.com (U.-H.P.); cj.ko@edgc.com (C.J.G.); dabin.park@edgc.com (D.P.); yg.lim@edgc.com (Y.G.L.); ks.lee@edgc.com (I.K.L.); woojong.do@edgc.com (W.-J.D.); kyoungjoo.lee@edgc.com (K.J.L.); hj.kim@edgc.com (H.K.); nayoung.min@edgc.com (N.Y.M.);; 2Department of Computer Science and Engineering, Incheon National University (INU), Incheon 22012, Republic of Korea; 3NGENI Foundation, San Diego, CA 92123, USA; 4Division of Pulmonary and Critical Care Medicine, Department of Medicine, Samsung Medical Center, Sungkyunkwan University School of Medicine, 81 Irwon-ro, Gangnam-gu, Seoul 06351, Republic of Korea; sangwonum@skku.edu; 5Diagnomics, Inc., 5795 Kearny Villa Rd., San Diego, CA 92123, USA

**Keywords:** liquid biopsy, multi-omics, machine learning, lung cancer, cell-free DNA, copy number variation, non-invasive, tumor marker, genomics

## Abstract

**Simple Summary:**

Early lung cancer detection is vital. Next-generation sequencing (NGS) enables cell-free DNA (cfDNA) liquid biopsy to detect genetic changes, such as copy number variations (CNVs). Recent machine learning (ML) analyses using cancer markers can identify anomalies, and developing methods based on ML for patient big data analysis is crucial for predicting cancer. We analyzed blood samples of 92 lung cancer patients and 80 healthy individuals, detecting significant differences in cancer markers, cfDNA concentrations, and CNV screening. Here, we used three algorithms of ML, such as Adaptive Boosting (AdaBoost), Multi-Layer Perceptron (MLP), and Logistic Regression (LR). ML analysis using cancer markers, cell-free DNA, and CNV individually exhibited relatively low discriminative power between cancer patients and healthy individuals. However, integrating multi-omics data into ML significantly improved accuracy, suggesting potential for precise cancer diagnosis. This study suggests the prospect of effectively distinguishing and diagnosing lung cancer from healthy individuals through blood-based ML analysis.

**Abstract:**

Early detection of lung cancer is crucial for patient survival and treatment. Recent advancements in next-generation sequencing (NGS) analysis enable cell-free DNA (cfDNA) liquid biopsy to detect changes, like chromosomal rearrangements, somatic mutations, and copy number variations (CNVs), in cancer. Machine learning (ML) analysis using cancer markers is a highly promising tool for identifying patterns and anomalies in cancers, making the development of ML-based analysis methods essential. We collected blood samples from 92 lung cancer patients and 80 healthy individuals to analyze the distinction between them. The detection of lung cancer markers Cyfra21 and carcinoembryonic antigen (CEA) in blood revealed significant differences between patients and controls. We performed machine learning analysis to obtain AUC values via Adaptive Boosting (AdaBoost), Multi-Layer Perceptron (MLP), and Logistic Regression (LR) using cancer markers, cfDNA concentrations, and CNV screening. Furthermore, combining the analysis of all multi-omics data for ML showed higher AUC values compared with analyzing each element separately, suggesting the potential for a highly accurate diagnosis of cancer. Overall, our results from ML analysis using multi-omics data obtained from blood demonstrate a remarkable ability of the model to distinguish between lung cancer and healthy individuals, highlighting the potential for a diagnostic model against lung cancer.

## 1. Introduction

Lung cancer is one of the leading causes of cancer-related deaths worldwide [1,2]. While numerous genetic and molecular alterations have been identified in lung cancer, the complex interplay of these alterations remains largely unknown. Early screening of lung cancer is important because it can lead to early detection and diagnosis of the disease, which can improve treatment outcomes and patient survival rates [2,3].

Analysis of liquid biopsies of cell-free DNA (cfDNA) is promising for non-invasive cancer genome analysis that has several potential applications in cancer diagnosis, monitoring, and treatment [4,5]. Small fragments of cfDNA are released into the bloodstream by cancer cells and can be detected and analyzed using NGS technology, making them advantageous for detecting cancer at an early stage [6,7,8]. Through the analysis of cfDNA in the bloodstream, cancer-specific genetic alterations, including somatic mutations, copy number variations, and chromosomal rearrangements, can be detected. These findings hold the potential to facilitate early stage cancer diagnosis [9].

New cancer diagnosis technologies, such as copy number variation (CNV), somatic mutation, and methylation analysis of the cancer genome using next-generation sequencing (NGS) technology hold great promise for improving cancer diagnosis and treatment outcomes [10,11]. NGS technology has revolutionized cancer genomics by enabling high-throughput analysis of the cancer genome, helping to identify genetic alterations and epigenetic modifications that are associated with cancer development and progression, which can be used to develop more effective detection and personalized cancer treatments [12,13].

CNVs are common in cancer and play a role in tumor development. They can arise from errors in DNA replication, recombination, and chromosome segregation [14,15]. CNV analysis involves detecting and quantifying copy numbers in cancer cells compared with normal cells, with techniques like fluorescence in situ hybridization, comparative genomic hybridization, and NGS being used. NGS is particularly valuable for high-resolution data on CNVs, detecting small-scale CNVs, and identifying CNVs in non-coding regions, which is crucial in cases with a mix of tumor and normal cells, like cell-free DNA analysis [16,17].

Accurate early stage cancer diagnosis requires integrating multiple screening results. Cancer is a complex disease involving genetic and epigenetic changes across various biological levels [18]. Multi-omics analysis has shown great potential in improving diagnosis, predicting treatment responses, and discovering therapeutic targets. However, dealing with large and complex multi-omics datasets poses challenges for traditional statistical methods in uncovering intricate healthcare data relationships [19]. Moreover, the combination of multi-serum tumor markers has the potential to amplify the sensitivity of the detection methodology [20]. This augmentation is particularly advantageous in the context of early disease detection, contributing to the overall enhancement of diagnostic procedures’ efficacies.

Hence, machine learning (ML) plays an important role in digital healthcare for the early detection, prediction, and treatment of medical diseases. By utilizing large amounts of data, machine learning algorithms can detect patterns and anomalies that may indicate the presence of a disease at an early stage and also analyze patient data to predict the likelihood of developing a disease, rendering it as a valuable diagnostic tool [21,22]. Various ML algorithms, such as logistic regression (LR), random forest (RF), support vector machine (SVM), and neural networks, can be used in disease diagnosis, medical imaging analysis, drug discoveries, and personalized treatments [23,24]. In a recent study, the potential of methylation-based cancer diagnosis using deep neural networks (DNNs) has been emphasized. This approach shows promise in effectively classifying cancer and normal samples, while also accurately identifying the tissue origin of cancer [25]. The predictive and differentiating capabilities of machine learning could play a pivotal role in improving patient survival rates and enabling early cancer detection, personalized treatments, predicting treatment responses, and enhancing treatment efficiency.

In this study, various omics data from lung cancer patients and healthy control individuals were collected and integrated with multi-omics analysis of blood samples in order to obtain a more comprehensive understanding of the mechanisms underlying lung cancer. Machine learning algorithms, such as Adaptive Boosting (AdaBoost), Multi-Layer Perceptron (MLP), and Logistic Regression (LR), were then applied to the integrated data to develop predictive models that could aid in diagnosis and early detection. Our results indicated that the integration of multi-omics data obtained from blood samples using machine learning algorithms has great potential for improving the diagnosis and early detection of lung cancer patients. By combining different types of data, such as CNVs from cell-free DNA, multiple protein biomarkers, and cell-free DNA concentrations, it is possible to develop more accurate and comprehensive models for cancer classification using non-invasive liquid biopsy approaches.

## 2. Materials and Methods

### 2.1. Patients and IRB

This study retrospectively analyzed 92 patients diagnosed with lung cancer and 80 healthy controls at Samsung Seoul Hospital from December 2020 to December 2021. The healthy control group consisted of subjects who were regarded as healthy according to the guidelines of the Gangnam Major Hospital Genome Project. Data on gender, age, tumor marker test results, TNM stage, and pathological findings in lung cancer patients were recorded. This study was conducted with the approval of the Samsung Seoul Hospital Research Ethics Committee (IRB No. 2018-01-081) and the Gangnam Major Hospital Genome Project Research Ethics Committee (IRB No. DR_CPLX_001).

### 2.2. Sample Preparation and cfDNA Extraction

This study enrolled 92 patients with lung cancer ranging from stage I to stage IV, as well as 80 healthy controls. A total of 10 mL of blood was collected from both cancer patients and healthy donors using Vangenes Cell-Free DNA Tubes (Vangenes, Torrance, CA, USA) centrifuged at 1900 g for 15 min at room temperature. cfDNA was isolated from 4 mL plasma by using Chemagic 360 (Perkin Elmer, Waltham, MA, USA) under the Chemagic Circulating NA 4K 360 H24 protocol. The quantification of cfDNA was performed using Qubit 2.0 using a dsDNA HS Assay Kit (Life Technologies, Carlsbad, CA, USA), which was used as input data for the S-can test.

### 2.3. Library Preparation and NGS Sequencing

A total of 5 ng cfDNA was used for library preparation with an NEBNext Ultra II DNA Library Prep Kit for Illumina (New England Biolabs, Ipswich, MA, USA) according to the manufacturer’s protocol. The library fragment size was determined by the 4200 TapeStation (Agilent Technologies, CA, USA) using High-Sensitivity D1000 ScreenTape (Agilent Technologies, Santa Clara, CA, USA). The libraries were then pooled and sequenced on an Illumina NextSeq Dx (Illumina, San Diego, CA, USA) with 75 cycles high-output kit with paired-end reads.

### 2.4. Serum Cancer Protein Marker Tests

The measurements for immunoassay testing were performed on cobas e602 (Roche, Basel, Switzerland) with Cyfra 21-1, CA 15-3, AFP, CEA, and CA 19-9 markers using 500 μL of plasma according to the manufacturer’s protocol. Statistical analysis was performed to compare lung cancer patients and healthy controls. The *p*-values of both groups were considered significant if they were less than 0.05.

### 2.5. Copy Number Profiling from Cell-Free DNA

Raw reads were mapped to the human reference genome (GRCh37/hg19) using BWA [26]. SAMtools (v1.4.1) [27] was used to sort and index bam files, removing those with a mapping quality below 1. After dividing the whole normalized genome into 10 Mb bins, the change in chromosome number was called using WisecondorX (v1.1.2) [28]. The detected CNV was separated into two classes, copy gain and copy loss, while several CNV regions from each chromosome were merged. For analysis, a total of 44 features were created by combining information from the two classes on the 22 autosomes.

### 2.6. Nested Cross-Validation

Machine learning analysis using multi-omics data followed the process shown in Figure 1. First, five tumor markers (Cyfra 21-1, CA15-3, AFP, CEA, and CA19-9), cfDNA extraction amount, and CNV data were used as input features. To enhance classification accuracy and mitigate overfitting, we utilized nested cross-validation, as previously described [29]. This approach involves an extended version of standard cross-validation, where each initial dataset partition (outer fold), comprising the training set, is subdivided into nested (inner fold) training and validation sets. Nested cross-validation was performed through a combination of outer cross-validation and inner cross-validation to evaluate the model’s predictive performance. The optimal hyperparameter value is determined in the inner cross-validation when the validation set is evaluated using a 5-fold cross-validation on the training set of each fold of the outer cross-validation. In the outer cross-validation, prediction and performance evaluation are performed on the test dataset with the parameters found in the inner cross-validation.

### 2.7. Machine Learning

All classifiers for the machine learning algorithm described in this study were built in the Python 3.5 environment and performed using the Scikit-learn library [30]. AdaBoost is a powerful ensemble classification method, combining weak classifiers into a strong one by dynamically adjusting sample weights. MLP is a feed-forward neural network with hidden layers, commonly trained using back-propagation for classification and regression tasks. LR is a popular and interpretable supervised learning algorithm that estimates class probabilities, making it effective for medical diagnosis, including cancer prediction from labeled patient data [31,32,33].

## 3. Results

### 3.1. Tumor Markers Are Significantly Increased in the Blood of Lung Cancer Patients

To determine whether the expression of cancer markers in cancer patients differed from that in healthy individuals (Table 1), we analyzed the protein levels in the blood. Protein markers were measured and analyzed to distinguish lung cancer from health control cases. Markers included CA 125 for ovarian cancer, Cyfra21-1 for lung cancer, CA15-3 for breast cancer, CA19-9 for pancreatic cancer, carcinoembryonic antigen (CEA) for colorectal, pancreatic, breast, and lung cancer, and alpha-fetoprotein (AFP) for liver cancer [34,35,36]. Lung cancer patients had significantly higher levels of Cyfra21-1, CA15-3, CEA, and CA19-9 compared with healthy controls (Table 2). The AFP level was not significant. The observed significant increase in cancer markers in patients provides evidence of their potential for distinguishing them from healthy individuals.

### 3.2. Low Specificity Observed in Machine Learning (ML) Analysis Using Individual Cancer Marker Expression

To identify the lung cancer diagnostic performance of each tumor marker, we performed five-fold cross-validation in three machine learning models using lung cancer patients and healthy controls. The ROC curve results are presented in Figure 2. For each marker, the AUCs for the AdaBoost, MLP, and LR models were as follows: Cyfra21-1 (0.734, 0.756, and 0.781), CA15-3 (0.555, 0.603, and 0.612), AFP (0.482, 0.510, and 0.484), CEA (0.713, 0.797, and 0.791), and CA19-9 (0.570, 0.558, and 0.556) (Figure 2, Table 3). Among markers Cyfra21-1 and CEA had higher AUC values, followed by CA15-3, CA19-9, and AFP.

ROC curve analysis was conducted on the lung cancer diagnostic performance using a combination of lung cancer-related markers, and the AUC values of the three machine learning models are presented in Table 3. With the AdaBoost model, the combined use of Cyfra21-1 and CEA resulted in the highest AUC of 0.828, which was statistically higher than the AUCs of 0.734 and 0.713 when using Cyfra21-1 or CEA individually, respectively. For the MLP model, the combination of Cyfra21 and CEA also yielded the highest AUC of 0.821, which was statistically significantly higher than the AUCs of 0.756 and 0.797 when using Cyfra21-1 or CEA individually, respectively (*p* < 0.05). With the LR model, the combined use of Cyfra21-1 and CEA also resulted in the highest AUC of 0.821, outperforming the AUCs of 0.781 and 0.791 when using Cyfra21-1 or CEA individually, respectively (*p* < 0.05).

### 3.3. The ML Analysis Utilizing the Concentration of Extracted cfDNA and the CNV Score Demonstrated a Notably High Level of Specificity

We conducted machine learning analysis to investigate whether lung cancer patients could be distinguished from healthy individuals using cfDNA concentration and CNV score as features. In the ROC curve analysis, the blood cfDNA concentration had AUC values of 0.706, 0.760, and 0.736 for AdaBoost, MLP, and LR models. For CNV, the corresponding AUC values were 0.856, 0.921, and 0.835 (Figure 3). No significant differences in analysis were observed among the three ML methods. These results demonstrate that cancer patients and healthy individuals can be distinguished using the cfDNA concentration and CNV score as features.

### 3.4. Selection of the Best Combination of Multi-Omics Data for Lung Cancer Diagnosis

In order to enhance the accuracy of lung cancer diagnosis through the utilization of multi-omics data, we combined protein markers, cfDNA concentration + CNV screening, and all markers. Among the tumor markers, Cyfra21-1, CEA, and CA15-3 exhibited a statistically significant difference between lung cancer patients and healthy controls, but only Cyfra21-1 and CEA were used for subsequent analysis, as they are commonly used for lung cancer diagnosis. Clinical studies have shown that lung cancer patients with high serum levels of Cyfra21-1 and CEA in the advanced stages (III and IV) exhibit shorter survival rates and are deemed to be good candidates for adjuvant chemotherapy [37,38].

This study assessed the performance of each machine learning model when using a single data type and compared the results with a multi-omics approach, which are summarized in Table 3 and Figure 4. Each sensitivity value was calculated with a fixed specificity of 90%, making it easier to assess the trade-off between sensitivity and specificity, which is vital in determining the overall accuracy of a test. When analyzing solely the cfDNA concentration, the AUC results from AdaBoost, MLP, and LR were 0.706, 0.760, and 0.736, respectively (Figure 3). When utilizing solely CNV data, the AUC results from AdaBoost, MLP, and LR were 0.856, 0.921, and 0.835, respectively (Figure 3). The AUCs of each model when using the combination of all three omics were 0.914, 0.931, and 0.914, with the MLP model producing the highest AUC (Figure 4, green line). Furthermore, we evaluated whether the use of cancer score values was effective in distinguishing between cancer patients and healthy individuals. A substantial difference in cancer scores between the normal and cancer groups suggested an enhanced ability to classify these two groups (Figure 5). Notably, the analysis of multi-omics data revealed that the combination of all multi-omics data yielded the most distinct separation between the cancer scores of the two groups (Figure 5D). Overall, when comparing the performance across all omics combinations, the AUC using all multi-omics data yielded the highest value.

### 3.5. Comparison of Detection Probabilities for Lung Cancer Patients by Stage Using Multi-Omics Data

While identifying cancer in the early stages is essential for identifying appropriate treatment solutions, assessing the cancer detection performance at each stage is a critical factor in determining a patient’s prognosis and treatment options. Incorporating blood cancer markers (Cyfra21-1 + CEA) into the three ML analyses exhibited varying AUC values across different stages. The AUC for stage I was poor, hovering around 0.6, whereas stages II and III exhibited substantial improvement, reaching around 0.8. Remarkably, stage IV consistently exceeded 0.9, indicating a notably high level of accuracy. These findings highlight the efficacy of ML utilizing blood cancer markers, particularly for identifying cancer patients in high-risk stage IV (Figure 6A).

The analysis of the cfDNA concentration across different lung cancer stages consistently revealed trends among models (AdaBoost, MLP, and LR). The AUC values for stages I, II, and III ranged from 0.626 to 0.845, indicating relatively modest performances. Even for stage IV, the AUC values ranged from 0.729 to 0.811 (Figure 6B). In terms of the CNV score, stage I exhibited AUC values between 0.752 and 0.876 across models. For higher-risk stages II and III, the AUC values varied from 0.768 to 0.880. Remarkably, stage IV displayed an AUC exceeding 0.9, demonstrating strong discriminatory power (Figure 6C). These findings underscore the potential of the cfDNA concentration and CNV score in distinguishing cancer stages.

The analysis based on the integration of all multi-omics data for different cancer stages demonstrated remarkably high AUC values in the machine learning analysis. The three ML models (AdaBoost, MLP, and LR) displayed values around 0.8 for stage I (0.797, 0.874, and 0.861), and over 0.9 for stages II (0.938, 0.964, and 0.875) and III (0.900, 0.903, and 0.889). Notably, for stage IV (0.983, 0.966, and 0.966), AUC values exceeding 0.966 were observed, indicating an exceptionally high level of discrimination (Figure 6D). Appendix A shows the overall sensitivity at the 95% confidence interval of each machine learning model in different stages of lung cancer. The average sensitivity percentages for ML models, calculated at a specificity of 90%, were as follows: stage 1 (50.0%, 63.6%, and 68.2%), stage 2 (80.0%, 90.0%, and 80.0%), stage 3 (69.6%, 73.9%, and 65.2%), and stage 4 (96.6%, 93.1%, and 89.7%), with the highest sensitivity observed for stage 4. Overall, these results underscore the significant effectiveness of integrated multi-omics data in predicting the stages of lung cancer patients, enhancing the potential for early cancer detection and the development of potential screening methods.

In summary, Figure 7 displays the AUC values for the single and combined multi-omics data in AdaBoost, emphasizing the performance improvement achieved with multi-omics machine learning analysis, showcasing an AUC of 0.914 and AUC of 0.986 for all-stage and stage IV patients, respectively.

## 4. Discussion

Lung cancer is a serious disease that is associated with high morbidity and mortality rates [39]. It is the leading cause of cancer-related deaths worldwide, accounting for approximately 1.8 million deaths each year [40]. Lung cancer is often asymptomatic in the early stages, and by the time symptoms develop, the disease may have progressed to an advanced stage, which can be difficult to treat [41]. As a result, early detection and diagnosis of lung cancer are critical for improving treatment outcomes and patient survival rates.

The current widely used screening test for lung cancer Is low-dose computed tomography (LDCT). While LDCT has the benefit of being quick and painless, the scan can produce a high rate of false positives that can lead to unnecessary follow-up procedures and, in some cases, early stage lung cancer may not even be detected by the scan [42,43]. In this study, we were able to accurately classify lung cancer patients and healthy controls based on a machine learning model using multi-omics data. We observed that the prediction probability was modest in the detection method using a single blood marker, but the probability was increased in the detection method of multi-blood markers [20]. This led to the discovery that accuracy calculations yielded significantly higher detection probabilities. The multi-omics data combination consisted of blood tumor markers, the cell-free DNA (cfDNA) extraction amount, and DNA copy number variation (CNV) data from 92 lung cancer patients and 80 healthy control samples and analysis was performed using AdaBoost, MLP, and LR machine learning models. When utilizing Cyfra21-1 and CEA, recognized as lung cancer-associated tumor markers in Adaboost, the AUC values were 0.734 and 0.713, respectively (Table 3). However, their combined application resulted in a notably increased AUC value of 0.828. Similar results were observed in the machine learning analysis of MLP and LR (Table 3). In the ML analysis utilizing solely cfDNA concentration data and CNV screening, the AdaBoost algorithm yielded AUC values of 0.706 and 0.856, respectively (Figure 3A). The combined analysis of ML in the cfDNA concentration and CNV screening yielded an AUC value of 0.880 in AdaBoost (Figure 4A red line). When all three combinations of multi-omics were used, the AUC value was 0.914 in AdaBoost (Figure 4A green line), which was the highest and exhibited the best classification performance among the other omics combinations. Similar results were observed in the machine learning analysis of LR (Figure 4C). In the context of the MLP algorithm, the AUC value for CNV alone was 0.921 (Figure 3B), surpassing the AUC value of 0.903 achieved by combining the cfDNA concentration and CNV screening (Figure 4B red line). However, when all multi-omics factors were integrated, the AUC value reached 0.931 (Figure 4B green line), representing the highest performance. The integration of protein markers, cfDNA concentration, and CNV screening through the ML algorithm led to an increased AUC value, indicating the potential for a diagnostic model against lung cancer.

ML analysis techniques have enhanced the integration of multi-omics approaches by enabling the comprehensive interpretation of large and complex datasets, which can be difficult to analyze using traditional statistical methods [44]. For example, by using machine learning algorithms, it may be possible to identify patterns of CNV, tumor markers, and cfDNA concentrations that are associated with a specific cancer type, stage, and individual patient, and use this information to develop a predictive model for cancer diagnosis and treatment [45]. AUC is a commonly used performance metric in ML to evaluate the accuracy of a classification model. Our high AUC values, namely 0.914 in AdaBoost, 0.931 in MLP, and 0.914 in LP, obtained through the comprehensive integration of all multi-omics data demonstrate the exceptional discriminatory ability of the model in distinguishing between lung cancer patients and healthy individuals. Moreover, the application of combined multi-omics data revealed notably elevated AUC values in stage IV (0.983, 0.966, and 0.966 for AdaBoost, MLP, and LP respectively), indicating the efficacy of our analytical approach in stage-specific prediction and discrimination. These results suggest the potential for a diagnostic model against lung cancer in oncological research.

The use of various omics technologies, such as somatic mutation detection, CNV, proteomic and epigenetic analysis using NGS, and array technology, holds great promise for improving the accuracy of cancer detection and diagnosis [18]. It is also important to note that the results of multi-omics cancer analysis using machine learning for early diagnosis can potentially be applied to other types of cancer diagnosis and early detection, as changes in CNV, the protein tumor marker levels, and cfDNA concentration are commonly observed in many types of cancer. However, each type of cancer is unique, and cancer is a complex disease that involves genetic and epigenetic changes, which can be reflected in the CNV of certain genes. Additionally, many types of cancer are associated with changes in the levels of specific protein biomarkers in the bloodstream, and the level can be affected by other non-cancer factors [46]. Furthermore, the cfDNA concentration can be altered in many types of cancer, and the concentration of cfDNA can be also affected by inflammation and infection [47]. Recently, characteristics of cfDNA, such as fragment size, preferred ends, end motifs, and single-stranded jagged ends, have been utilized in the field of cancer diagnosis. The levels of reduction in short fragment size, preferred ends, and the end motif “CCCA” are being examined for their potential in differentiating patients with hepatocellular carcinoma and healthy individuals. Further analysis of these characteristics of cfDNA is likely to enhance the probabilities of predicting and distinguishing cancer with even greater accuracy [48,49,50]. Therefore, it is important to note that the optimal biomarkers and machine learning models for each type of cancer may vary. Further research and validation of multi-omics approaches using machine learning algorithms will be necessary to fully realize the potential for multi-cancer diagnosis and early detection.

While our results exhibited a remarkably high accuracy in distinguishing cancer patients, it can be argued that the sample size representing lung cancer patients was too small. Due to the limited scale of the patient group, consisting of cases of lung cancer, we iteratively examined whether various detection methods, such as blood markers, cfDNA, and CNV, yielded similar outcomes for each approach. Given the modest patient group size, we repeatedly verified these analyses. To clarify whether these research findings hold true in a larger population, an increase in external validation data through the expansion of the patient group is required to obtain more accurate and meaningful results. To solve the matter of a small cohort, we are currently conducting further research that uses an enlarged patient cohort for investigation.

Despite its numerous advantages, multi-omics cancer analysis has some limitations that should be taken into consideration. Integrating data from different omics technologies is a complex task that requires specialized knowledge and expertise, and the process may be hindered by issues such as data quality, batch effects, and differences in technology platforms [51]. In addition, multi-omics analysis involves analyzing large amounts of data, which can be computationally intensive and require high-performance computing resources [52].

## 5. Conclusions

Overall, the integration of multi-omics cancer analysis approaches in liquid biopsy using machine learning techniques has great potential in significantly improving cancer diagnosis and treatment, which may lead to the development of more effective and personalized non-invasive cancer therapies, predicting treatment responses, and enhancing treatment efficiency, as well as early detection. The continual development and validation of robust predictive models and usage of additional biomarkers can present more meaningful diagnostics and treatment for a wide variety of cancer types.

## Figures and Tables

**Figure 1 cancers-15-04556-f001:**
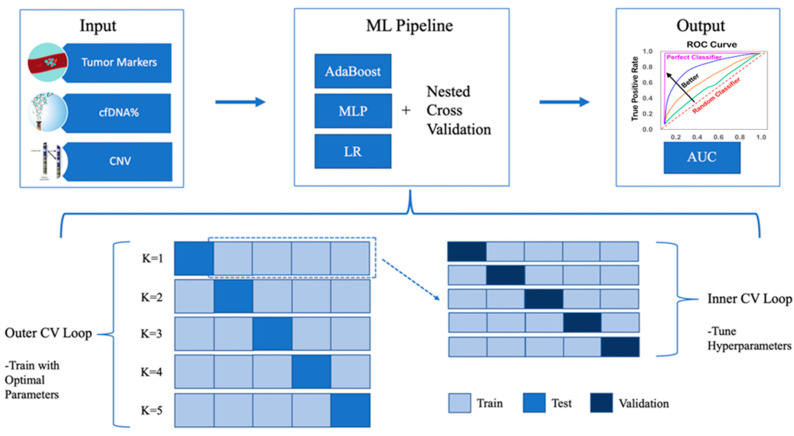
Overview of ML analysis using multi-omics data in accurately classifying lung cancer patients. The schematic illustrates the process of machine learning analysis using multi-omics data. Each square at the bottom represents each of the data sets analyzed in this study, with the training set shown in light blue, the test set in blue, and the validation set in dark blue.

**Figure 2 cancers-15-04556-f002:**
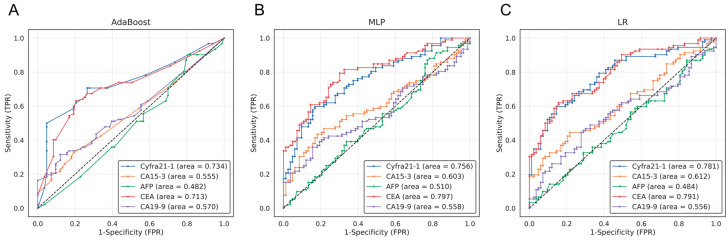
ROC curves based on machine learning model in lung cancer-screening tumor markers. The diagnostic performance validation results for the five tumor markers (AFP, CA15-3, CA19-9, CEA, and Cyfra21-1) using machine learning models ((**A**), AdaBoost; (**B**), MLP; (**C**), LR) are shown as ROC curves. The word “area” represents the ‘area under the curve (AUC)’. The x-axis and y-axis indicate the false positive rate (FPR) and true positive rate (TPR), respectively. The performance of the Cyfra21-1 tumor marker is represented by blue lines, while the CA15-3 marker is shown with orange lines. The AFP marker results are displayed in green, CEA in red, and CA19-9 in purple. Dashed line represents Random prediction.

**Figure 3 cancers-15-04556-f003:**
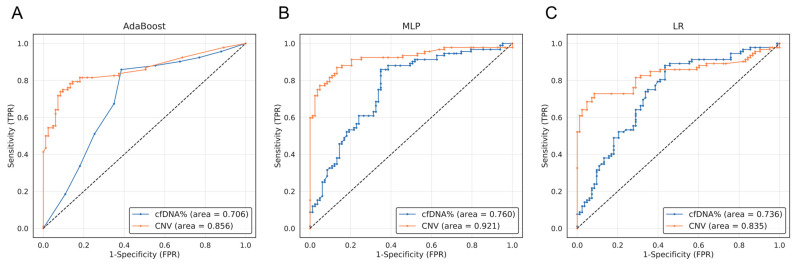
ROC curve analyzing blood cfDNA concentration and CNV based on machine learning models. ROC curves are presented to compare the blood cfDNA concentration and CNV between lung cancer patients and healthy individuals using three machine learning models: (**A**) AdaBoost, (**B**) MLP, and (**C**) LR. The CNV results are shown as orange lines, while the cfDNA concentration is represented by blue lines. Refer to the legend of Figure 2 for further information. Dashed line represents Random prediction.

**Figure 4 cancers-15-04556-f004:**
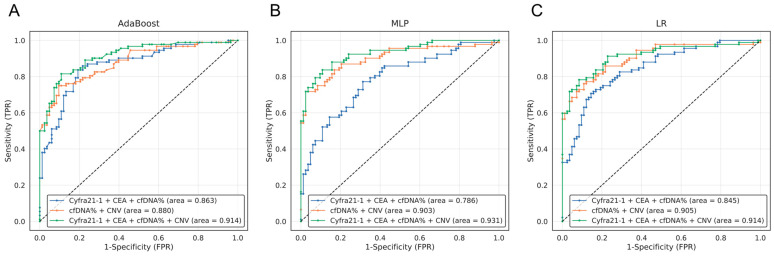
ROC curve based on a machine learning model using lung cancer multi-omics data. The performance of lung cancer diagnosis using combinations of multi-omics data was calculated and presented using the ROC curve. The results based on the machine learning models of AdaBoost, MLP, and LR are shown in (**A**–**C**). The blue lines represent the utilization of the markers Cyfra21-1, CEA, and concentration of cfDNA. The combination of cfDNA% and CNV is indicated by orange lines, and green indicates the combination of the four data sets. See the legend of Figure 2 for further information. Dashed line represents Random prediction.

**Figure 5 cancers-15-04556-f005:**
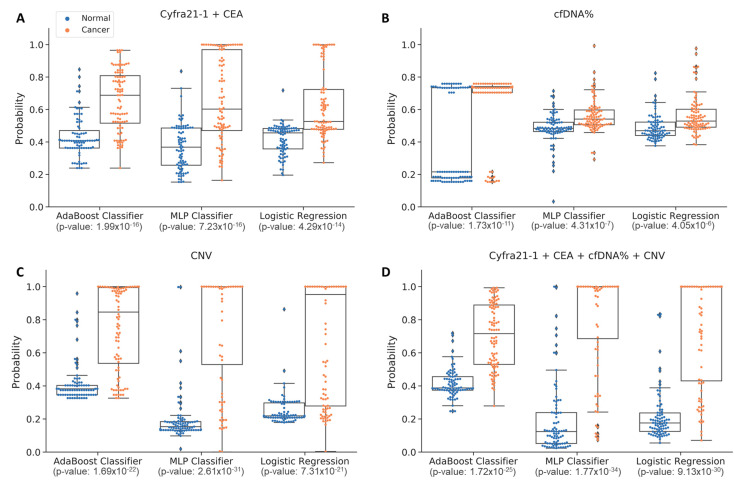
Comparison of cancer scores for each machine learning model in each data set. The cancer scores of various data sets using three machine learning models (AdaBoost, MLP, and LR) are represented by whisker plots. Blue dots show the calculation results of normal groups, and scores from lung cancer patients are indicated by orange dots. The x-axis represents the machine learning model used, while the AUC values are shown on the y-axis. (**A**) shows cancer scores using tumor markers (Cyfra21-1 + CEA), and (**B**) is the result of analysis using cfDNA%. Cancer scores based on CNV are represented in (**C**), and the results from the four data sets are shown in (**D**).

**Figure 6 cancers-15-04556-f006:**
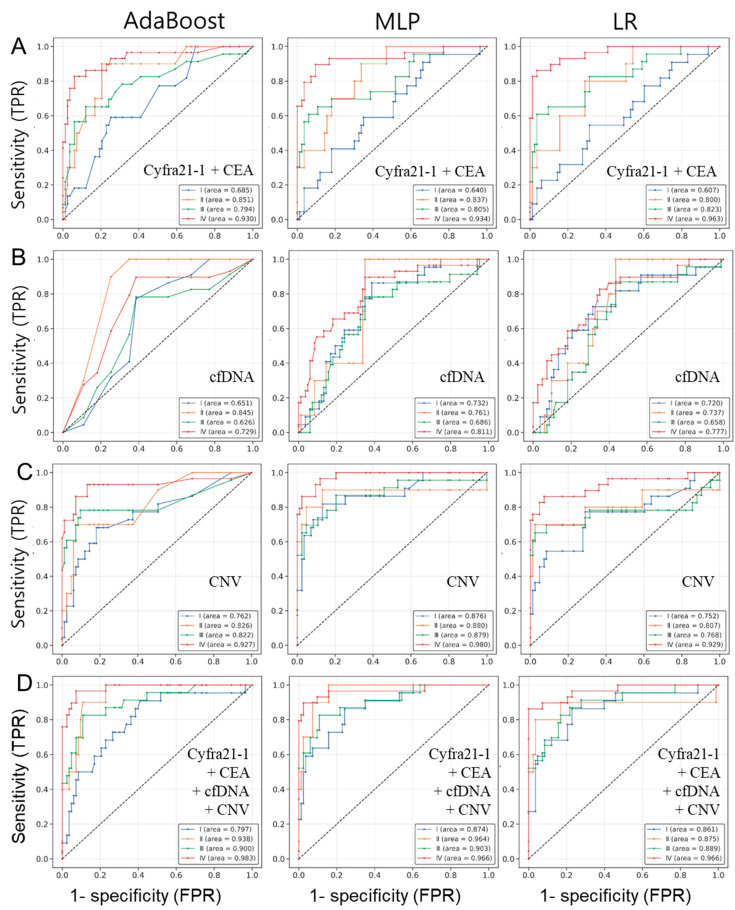
ROC curve based on a machine learning model using multi-omics data for the stage of lung cancer. The performance of diagnosis using blood marker (Cyfra21-1+CEA) (**A**), cfDNA concentration (**B**), CNV (**C**), and combinations of multi-omics data (Cyfra21-1+CEA+cfDNA+CNV) (**D**) was calculated and presented using the ROC curve for the stages of lung cancer. Results based on the machine learning models AdaBoost, MLP, and LR are shown. The blue line represents stage I, the orange line represents stage II, the green line represents stage III, and the red line represents stage IV. The analysis of ROC curves for single or multi-omics data was represented in each figure.

**Figure 7 cancers-15-04556-f007:**
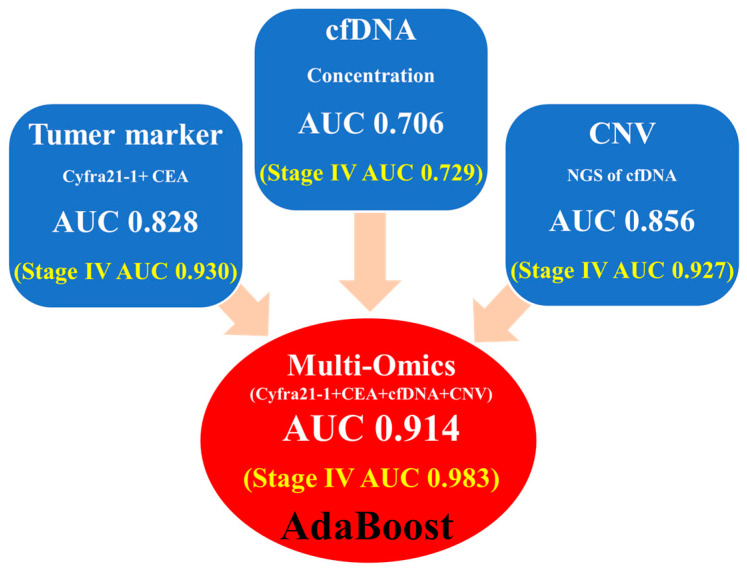
Overall average AUC value of machine learning performance. AUC values obtained from various data types are displayed in blue boxes. The values for tumor markers, CNV, and cfDNA concentration analyses are represented separately. Ultimately, the AUC value obtained from the multi-omics approach utilizing all three data types is shown in a red circle.

**Table 1 cancers-15-04556-t001:** Characteristics for participants with patients of lung cancer and healthy individuals.

Characteristics		Normal	Lung Cancer
**Enrollment**		80	92
**Gender**	Male	43	65
	Female	37	27
**Age**	Median	32	65
	Range	18–72	40–80
**Stage**	I	-	30
	II	-	10
	III	-	23
	IV	-	29
**Cancer Type**	Adenocarcinoma	-	56
	Squamous cell carcinoma	-	22
	Large-cell neuroendocrine carcinoma	-	1
	Non-small-cell lung cancer	-	7
	Small-cell lung cancer	-	2
	Not specified	-	2

**Table 2 cancers-15-04556-t002:** Measurement of tumor markers in patients with lung cancer and healthy subjects.

Protein Marker	Lung Cancer		Healthy		*p*-Value
	n	Mean ± SEM	n	Mean ± SEM	
Cyfra_21-1 (ng/mL)	92	5.87 ± 1.14	80	1.4 ± 0.09	0.0000
CA_15-3 (U/mL)	92	18.65 ± 3.07	80	8.68 ± 0.54	0.0009
AFP (ng/mL)	92	6.9 ± 4.74	80	2.27 ± 0.12	0.1659
CEA (ng/mL)	92	42.75 ± 14.97	80	1.84 ± 0.15	0.0037
CA_19-9 (U/mL)	92	40.21 ± 13.75	80	9.06 ± 0.64	0.0130

**Table 3 cancers-15-04556-t003:** Performance of machine learning models evaluated with a combination of Cyfra21-1 and CEA, known lung cancer markers, and other tumor markers.

Tumor Makers	AdaBoost	MLP	LR
Cyfra21-1	0.734	0.756	0.781
CA15-3	0.555	0.603	0.612
AFP	0.482	0.510	0.484
CEA	0.713	0.797	0.791
CA19-9	0.570	0.558	0.556
Cyfra21-1 + CEA	0.828 ^a,b^	0.821 ^a,b^	0.821 ^a,b^

^a^ *p* < 0.05 compares with Cyfra21-1, ^b^
*p* < 0.05 compared with CEA.

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
