# Peer review of "Enhancing Lung Cancer Classification through Integration of Liquid Biopsy Multi-Omics Data with Machine Learning Techniques"

_cancers, 2023, doi:10.3390/cancers15184556_

Round 1
Reviewer 1 Report
The authors describe a cohort study based on 92 patients with lung cancer and 80 healthy individuals. The assess the performance of lung cancer detection in different stages combining tumormarkers, cfDNA CNV and cfDNA concentration. Three different ML approaches are evaluated and compared. It is known that the combination of different diagnostic modelities will improve performance of detection.
Major comments:
1. I find the overall message of the paper a bit difficult to grasp, the authors seem to have two aims, a clinical one: detection of lung cancer, and two: the impact and comparison of different ML approaches and their performance. The latter seems to be the main message, leading to possibilities in the first one. While reading the paper the data and research is more on the different ML approaches and not on the clinical impact of the multi-omics test. This should be made clearer throughout the text.
2. In the introduction I am missing references to other studies on multi-omics (from combinations of tumor markers to more complex combinations) and some text on what this study adds to the current literature. Combinations of biomarkers and ML approaches have been evaluated (for example for tumormarkers (recent paper van Delft et al., tumor biology 2023)
3. Performance should be compared over the different stages, so include the performance of the different ML approaches (or the best) for each disease stage
4. Figure 6 should include the specificity at which these sensitivities are reached
5. The authors should elaborate on the impact of their relatively small cohort on the presented results. In addition, it would be relevant to elaborate on other diagnostic data (fragmentation/end motifs/…) and how this could further improve performance.
6. Is it feasible to include an explanation of the observed differences of the different ML approaches?
Minor comments:
1. First word “Authors” in the discussion should be deleted.
Reviewer 2 Report
A few comments for the authors to address:
1. The dataset is composed of 172 patients (cite from line 106) and 44 features (cite from line 144), do you worry about overfitting? Can you show the train and test performance to demonstrate there's little to no overfitting
2. Line 269 to 271: Can you show the precision along with the sensitivity?
3. Can you show the p values of the coefficients of the LR model AND feature importances for other models?
